# Mediterranean Diet and Lifestyle Habits during Pregnancy: Is There an Association with Small for Gestational Age Infants? An Italian Single Centre Experience

**DOI:** 10.3390/nu13061941

**Published:** 2021-06-05

**Authors:** Rachele De Giuseppe, Manuela Bocchi, Silvia Maffoni, Elsa Del Bo, Federica Manzoni, Rosa Maria Cerbo, Debora Porri, Hellas Cena

**Affiliations:** 1Laboratory of Dietetics and Clinical Nutrition, Department of Public Health, Experimental and Forensic Medicine, University of Pavia, 27100 Pavia, Italy; rachele.degiuseppe@unipv.it (R.D.G.); debora.porri01@universitadipavia.it (D.P.); 2Department of Clinical, Surgical, Diagnostic and Pediatric Sciences, University of Pavia, 27100 Pavia, Italy; manub1910@gmail.com (M.B.); elsa.delbo@unipv.it (E.D.B.); 3Clinical Nutrition and Dietetics Operative Unit, Internal Medicine and Endocrinology, ICS Maugeri IRCCS, 27100 Pavia, Italy; silvia.maffoni@icsmaugeri.it; 4Health Promotion-Environmental Epidemiology Unit, Hygiene and Health Prevention Department, Health Protection Agency, 27100 Pavia, Italy; f.manzoni@smatteo.pv.it; 5Neonatal Unit and Neonatal Intensive Care Unit, Fondazione IRCCS Policlinico San Matteo, 27100 Pavia, Italy; rm.cerbo@smatteo.pv.it

**Keywords:** small-for-gestational-age infants, Mediterranean diet, lifestyle habits

## Abstract

Background. The small-for-gestational-age (SGA) in infants is related to an increased risk of developing Non-Communicable Diseases later in life. The Mediterranean diet (MD) is related to lower odds of being SGA. The study explored retrospectively the association between SGA, maternal MD adherence, lifestyle habits and other SGA risk factors during pregnancy. Methods. One hundred women (16–44 years) with a pregnancy at term were enrolled. Demographic data, parity, pre-gestational BMI, gestational weight gain, pregnancy-related diseases, and type of delivery were collected. The MD adherence (MEDI-LITE score ≥ 9), physical activity level, and smoking/alcohol consumption were registered. SGA neonates were diagnosed according to the neonatal growth curves. Results. Women were divided into “SGA group” vs. “non-SGA group”. The MD was adopted by 71% of women and its adherence was higher in the “non-SGA group” (*p* = 0.02). The prevalence of pregnancy-related diseases (gestational diabetes/pregnancy-induced hypertension) was higher in the “SGA group” (*p* = 0.01). The logistic regression showed that pregnancy-related diseases were the only independent risk factor for SGA. Conclusions. MD may indirectly reduce the risk of SGA since it prevents and exerts a positive effect on pregnancy-related diseases (e.g., gestational diabetes and hypertension). The small sample size of women in the SGA group of the study imposes a major limitation to the results and conclusions of this research, suggesting however that it is worthy of further investigation.

## 1. Introduction

Pregnancy is a critical period during which foetal development may be influenced by different factors including maternal diet, which significantly contributes to the intra-uterine environment and permanently influences an organism’s physiology and metabolism [1]. In particular, a precious time window to shape the baby’s health and to prevent acute and chronic diseases through correct nutritional choices includes the first 1000 days, from conception to the second year of life [2]. In this context, prior studies have shown the association of specific nutrients or food groups consumed during pregnancy with the improvement or the prevention of some mother and baby diseases [2,3,4].

The Mediterranean diet (MD) is recognized as a healthy and balanced dietary pattern [5,6,7]. Several epidemiological studies and clinical trials support its role during pregnancy for the prevention of some maternal and foetal pathologies; in particular, high adherence to MD has been associated with a lower risk of gestational diabetes mellitus, pregnancy-induced hypertension and preeclampsia, preterm delivery and obesity in the offspring [8,9,10,11,12,13]. Furthermore, a moderate level of physical activity performed during healthy pregnancy plays a key role in the improvement of mother and baby’s health and the prevention of some pregnancy-induced complications [14,15].

Birth weight is considered the main determinant of perinatal morbidity and mortality [16]. The concept of small-for-gestational-age (SGA) infants considers birth weight, gestational age and sex [16]. SGA is related to an increased risk of developing Non-Communicable Diseases (NCDs) later in life, including diabetes mellitus, obesity, cardiovascular disease, and psychiatric disorders [16]. Maternal risk factors associated with SGA can be short stature, low weight, Indian or Asian ethnicity, nulliparity, gestational diabetes mellitus or hypertension-induced pregnancy and maternal lifestyle habits such as habitual alcohol consumption, cigarette smoking and the adherence to unhealthy dietary patterns [16]. Moreover, it has been recently reported that the Mediterranean dietary pattern is related to lower odds of being SGA [17].

Therefore, based on these considerations, the present study aimed to explore retrospectively the association between SGA and maternal adherence to the MD and lifestyle habits (e.g., physical activity level, smoking and alcohol consumption) as well as other SGA risk factors during pregnancy in a small Italian cohort.

## 2. Materials and Methods

The present study is a retrospective population-based study examining maternal MD adherence and lifestyle habits during pregnancy along with demographic and clinical and maternofoetal outcomes.

One hundred women (age range 16–44 years) with a pregnancy at term (gestational age > 37 and <42 weeks) and attending the Obstetrics Unit of the Fondazione IRCCS Policlinico San Matteo, Pavia (Italy) were enrolled between April 2019 and July 2021.

The study was approved by the Human Ethics Committee of Fondazione IRCCS Policlinico S. Matteo of Pavia (Protocol number: 20180022618; 6/12/2018) and it was conducted according to the Good Clinical Practice guidelines.

### 2.1. Demographic and Clinical Data and Maternofoetal Outcomes

Demographic and anamnestic data such as age (years), nationality, educational status (e.g., primary school degree, high school degree, university degree), parity, height (cm), pre-gestational weight (Kg) and the weight gain during pregnancy (Kg) were collected from the medical records. The pre-gestational Body Mass Index (BMI) was then calculated (Kg/m^2^) (Figure 1).

Neonates weighing less than the 10th percentile, adjusted for gestational age at delivery and sex, were defined as SGA [18]. Women were also divided into the “SGA group” and “non-SGA” group (neonates appropriate for gestational age) that was adopted as the control group (Figure 1). Since there were only three neonates large for gestational age [18], they were included in the “non-SGA” group.

Other maternal and foetal outcomes were also collected from the medical records. In particular, (i) pregnancy-related diseases (e.g., no diseases, gestational diabetes mellitus, GDM, pregnancy-induced hypertension); (ii) onset of urinary or genital tract infections in the mother during pregnancy (e.g., yes, no); (iii) type of delivery (spontaneous delivery, vaginal operative delivery, caesarean section) (Figure 1).

### 2.2. Mediterranean Diet Adherence and Lifestyle Habits

The MD adherence and lifestyle habits, including smoking habit, alcohol consumption and physical activity level during pregnancy were investigated before the delivery, during the routine visit of the mother.

The MD adherence was assessed using the MEDI-LITE score that was obtained from a previously validated questionnaire [19]. The questionnaire investigates the frequency of consumption of nine classes of food, namely (i) fruit, (ii) vegetables, (iii) cereal grains, (iv) legumes, (v) fish and fish products, (vi) meat and meat products, (vii) dairy products, (viii) alcohol intake and (ix) olive oil [19] (Figure 1).

The questionnaire’s score ranged from 0 to 18 where the highest value corresponded to the highest MD adherence [19]. In brief, for fruit, vegetables, cereal grains, legumes and fish, a score of 2 was assigned to a high-frequency consumption, a score of 1 to a moderate frequency consumption and a score of 0 to a low-frequency consumption [19]. Conversely, for meat/meat products and dairy products, low-frequency consumption scored 2, moderate frequency consumption scored 1 and high-frequency consumption scored 0 [19]. For alcohol consumption, alcohol units (1 alcohol unit = 12 g of alcohol) were considered. Thus, a score of 0 was assigned to the consumption of more than 2 alcohol units/day; a score of 1 was assigned to the consumption of one alcohol unit/day and a score of 2 was assigned to the consumption of 2 alcohol units/d [19]. Moreover, women were also classified as women who usually never consumed alcohol, women who stopped consuming alcohol during pregnancy, women who did not stop consuming alcohol during pregnancy.

Subjects reporting a score at or higher than 9 at the MEDI-LITE questionnaire had a significantly increased possibility of being adherents at MD, as previously reported [19]. Furthermore, according to their nationality, women were divided into subjects belonging to the Mediterranean basin culture (women coming from Italy, Albania, Morocco and Israel) and women not belonging to the Mediterranean basin culture (women coming from Romania, Russian, Indonesian, Senegal, Ecuador, Ivory Coast).

Physical activity was investigated by using the International Physical Activity Questionnaire, short-form (IPAQ; short form) [20]. The appropriate adapted country/language-specific IPAQ version questionnaire was downloaded [21].

The metabolic equivalent of task (MET-min) per week was calculated as METs = MET level * minutes of activity * events per week [21] (Figure 1).

Physical activity level was also classified as sedentary (total METs < 699), moderate (total METs ranging 700–2519) and high (total METs > 2520) [20].

Smoking habit (e.g., women who never smoked, women who stopped smoking during pregnancy, women who did not stop smoking during pregnancy) and alcohol consumption (e.g., women who never consumed, women who stopped drinking alcohol during pregnancy, women who did not stop drinking alcohol during pregnancy) were also investigated.

### 2.3. Statistical Analysis

Basic description of data and statistical analyses were performed using MedCalc software (MedCalc software Ltd., Ostend, Belgium). Data were described as absolute numbers and percentages for categorical variables, while medians and interquartile ranges (IQR) were used to describe ordinal variables and numerical variables, which did not follow a normal distribution (tested with the Shapiro–Wilk test). The chi-square test was adopted for categorical variables, while the Kruskal–Wallis test (with the Mann–Whitney U test as posthoc test) and Spearman’s rank correlation test for ordinal and numerical variables in bivariate analysis. Additionally, a logistic regression model with SGA as an outcome variable was created. The predictor variables simultaneously entered in the model included age, BMI before pregnancy, gestational weight gain, pregnancy-related diseases, MD adherence, METs, alcohol consumption and smoking habit.

**Figure 1 nutrients-13-01941-f001:**
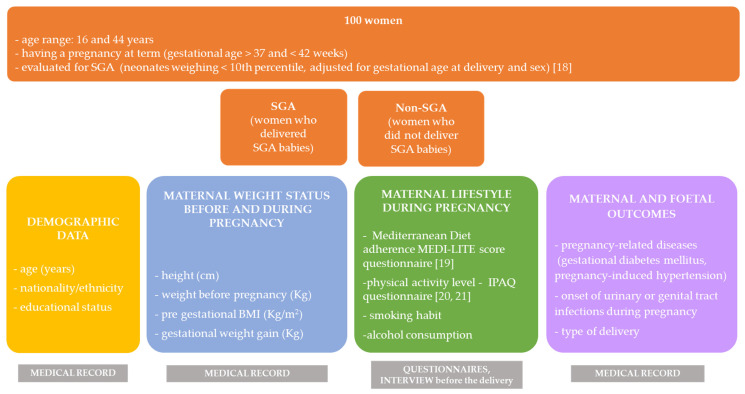
Study population and variables collected. Legend. BMI: body mass index; SGA: small for gestational age.

## 3. Results

Table 1 reported the demographic and anamnestic data, the maternal and foetal outcomes, the MD adherence and lifestyle habits of all women; data were also reported according to SGA that occurred in 10% of cases.

In brief, the total number of women enrolled in the present research consisted of 100 women (median age: 33 years; IQR: 29–36). The majority of women were Italian (81%), others coming from Albania (7%), Morocco (1%), Israel (1%), Romania (5%), Russian (1%), Indonesian (1%), Senegal (1%), Ecuador (1%), and Ivory Coast (1%). Furthermore, most of them (90%) came from regions with a Mediterranean culture including Italy, Albania, Morocco and Israel. The level of education was well represented in the three grades as 26% of women reported a low level of education (primary school), 41% an intermediate level of education (high school) and 33% a high level of education (university). As for demographic data, no significant differences were reported between the “SGA” and the “non-SGA” groups (Table 1).

Women gained a median of 13 Kg (IQR: 10.00–16.00 Kg) during pregnancy and even if they presented pre-gestational BMI within the range of normal weight status (median 22.45 Kg/m^2^; IQR 16.65–26.50 Kg/m^2^), an inverse and significant association between pre-gestational BMI and the educational level (−0.22; *p* = 0.03) was observed (Table 2).

Most women were nulliparous (54% of cases) or had only one child (34% of cases) and the spontaneous delivery occurred in 82% of cases. Moreover, parity was significantly higher in the “SGA group” than in the “non-SGA group” (median: 1.00; IQR:1.00–1.00 vs. median: 0.00; IQR:0.00–1.00; *p* = 0.03, Table 1). No urinary or genital tract infections were observed during pregnancy in the majority of women (75% of cases). Furthermore, although 83% of women had no pregnancy-related medical complications such as gestational diabetes (GDM) or pregnancy-induced hypertension, and the prevalence rate was higher in the “SGA group” when compared to the “non-SGA group” (gestational diabetes: 20% vs. 9.1%; pregnancy-induced hypertension: 20% vs. 3.40%; *p* = 0.01, Table 1).

The median value of the MEDI-LITE score was 11.00 (IQR: 9.00–13.00) and the MD was adopted by 71% of women. Particularly, the adherence to the Mediterranean model was significantly higher in the “non-SGA” group than in the “SGA group” (60% vs. 25.56%; *p* = 0.02) (Table 1). The MEDI-LITE score was positive and significantly associated with age (*r* = 0.22; *p* = 0.02, Table 2).

The majority of women had a moderate (40% of cases) or high physical activity level (38% of cases); the median METs value of the whole population was 1815.00 (IQR: 825.00–3390.00), with no differences among groups. A positive association was found between METs and parity (*r* = 0.22; *p* = 0.03, Table 2).

Most women never consumed (48% of cases) alcohol during pregnancy and 33% of cases stopped drinking alcohol during pregnancy (33% of cases); similarly, 68% of women never smoked during pregnancy while 20% stopped smoking during pregnancy (20% of cases). The Spearman’s rank correlation reported that women who never stopped drinking alcohol during pregnancy gained less weight (*r* = −0.20; *p* = 0.05, Table 2).

Last, logistic regression analysis (Table 3) was conducted to evaluate the association between SGA and maternal demographic, clinical and lifestyle variables. SGA as the dependent variable and other maternal demographic, clinical and lifestyle variables (e.g., age, level of education, pre-gestational BMI, gestational weight gain, pregnancy-related diseases, MD adherence, METs, smoking habit and alcohol consumption) as the independent ones, were simultaneously assessed. Since most women were nulliparous (54% of cases) or had only one child (34%), parity was not included in the model. The analysis showed that women who reported pregnancy-related diseases had a higher odds of having SGA babies than those who did not report pregnancy-related diseases (OR: 3.01; 95% CI: 1.01–8.95; *p* = 0.05, Table 3).

## 4. Discussion

Being small for gestational age is associated with increased risk of stillbirth and neonatal mortality, and the risk of SGA is related to socio-demographic maternal variables (age or socioeconomic status), maternal weight status, chronic diseases, (e.g., diabetes or hypertension), and nulliparity and maternal lifestyle risk factors (e.g., nutrition, level of physical activity, smoking and alcohol consumption) [22,23,24].

The present research retrospectively investigated the association between SGA and maternal adherence to the MD and lifestyle as well as other SGA risk factors during pregnancy in a small Italian cohort. In our sample, SGA was observed in 10% of women.

Several studies have shown that the high nutritional quality of MD prevents micronutrient deficiencies, so common and threatening not only the health of mothers but also programming and foetal development [22,25]. Similarly, it is well known that during gestation, essential nutrients are transferred from the maternal to foetal circulation via the placenta [22]. Indeed, MD is rich in monounsaturated and polyunsaturated fatty acids, micronutrients and bioactive molecules, appropriate to fulfil nutritional requirements and ensure adequate foetal growth and development [4,13,26]. Very recent findings also confirmed that the MD pattern is related to lower odds of being SGA [17]. Among maternal lifestyle factors, it is well known that adequate physical activity level during pregnancy is safe and can affect the pregnancy outcomes beneficially [27] as well as maternal alcohol consumption and smoking, which are salient predictors of pregnancy and foetal outcomes, including SGA [23]. Last, pre-gestational BMI and weight gain during gestation are predictors of foetal growth and neonatal outcomes [24].

The findings of this study confirmed that generally, women consider it important to improve their lifestyle during reproductive age and pregnancy and in our sample, this was demonstrated by the high adherence to the MD, characterized by plant-based foods high in fibre, as well as to the physical activity recommendations. Indeed, 71% and 68% of the total sample showed a high adherence to the MD and moderate or high levels of physical activity, during pregnancy. Specifically, exercise has the potential to maintain an adequate BMI and prevent excessive gestational weight gain and pregnancy-related complications, including gestational diabetes and hypertension [27]. In the present research, this was confirmed by the fact that: (i) the women had a median pre-gestational BMI within the normal weight range; (ii) women reported overall physiological gestational weight gain; (iii) the prevalence of gestational diabetes and hypertension was only 10 and 5%, respectively.

However, women with a lower level of education experienced a significantly higher pre-gestational BMI, as already demonstrated by others [28] reporting that the most substantial decrease in BMI was associated with an increase in income and education, especially among white women.

Again, most of the women never consumed or stopped drinking alcohol during pregnancy (81%), as well as never smoked or quit smoking (88%). The Spearman’s rank correlation described a significant and negative association between women who never stopped alcohol consumption during pregnancy and gestational weight gain. However, besides alcohol consumption, many other maternal socio-demographic and lifestyle factors may affect weight gain during pregnancy [29,30]; therefore, this association should be further reconsidered and investigated.

To better explore the association between demographic, clinical and lifestyle variables and SGA, the sample population was first divided into SGA group and non-SGA group. In general, no significant differences among the different variables were found between the groups except for the MD adherence, pregnancy-related diseases and parity.

Considering the adherence to the MD, although the median MEDI-LITE score did not differ between the non-SGA and the SGA group, the percentage of women adhering to the MD was significantly higher and about double in the non-SGA group than the SGA-group. This result was in agreement with previous findings that revealed that women with high adherence to the MD had a positive impact on pregnancy outcomes, especially on the foetal ones [17]. Additionally, the MEDI-LITE score was significantly and positively associated with age; this result could be linked to higher food and nutritional knowledge among older women that has not been investigated [31]. Again, parity was significantly higher in SGA groups and women of the non-SGA group were in median nulliparous. Our results were in contrast with previous findings [23]. Indeed, it is well-known that SGA is associated with nulliparity as a direct consequence of physiological conditions, supposing that each following pregnancy the body becomes more efficient [23]. As for pregnancy-related diseases, the percentage of women not developed gestational diabetes or pregnancy-induced hypertension was significantly higher in the non-SGA women than their counterparts confirming the role of gestational diabetes and hypertension as risk factors for neonatal complications, such as SGA [32,33].

Last, the logistic regression evaluated the association between SGA (the dependent variable) and the maternal demographic, clinical and lifestyle factors (the independent variables) analysis. Since most of the women (88%) were nulliparous or had only one child we did not consider parity in this model. The significant differences mentioned above were only partially maintained. Even if the low adherence to the MD was associated with SGA the result did not reach significance. On the contrary, pregnancy-related diseases were independent factors for SGA.

## 5. Strengths and Limitations of the Study

In this study, some limitations should be acknowledged. First of all, sample size, which might have prevented the statistical confirmation of some associations between lifestyle, diet and foetal–maternal outcomes, reported by other authors.

Then, other factors that could influence dietary choices, including behavioural–cognitive features as well as previous diseases, or risk factors that may have exposed women to foetal–maternal complications considered in this study.

Finally, the timing of survey administration—women were interviewed at the end of the pregnancy and therefore it is logical that they may have reported more representative responses of the last quarter than to the whole gestation.

However, data on lifestyle and diet were collected using validated questionnaires, throughout interviews conducted by a previously trained midwife, integrated with medical records information and considering both diet and physical activity as key components of lifestyle during pregnancy able to impact pregnancy outcome.

## 6. Conclusions

The Mediterranean diet is a healthy dietary pattern and should be adopted during pregnancy, and studies reported that adherence to the MD is associated with a reduced risk of SGA. In the present study, we investigated retrospectively the association between SGA and the maternal adherence to the MD, physical activity, smoking habit, alcohol consumption and other SGA risk factors during pregnancy in a small Italian cohort.

Even if in our cohort, the adherence to the MD was not an independent factor for the prevention of SGA we can conclude that MD may indirectly reduce the risk of SGA since it prevents and exerts a positive effect on pregnancy-related diseases such as gestational diabetes and hypertension [5].

We acknowledge that the small sample size of women in the SGA group of the study imposes a major limitation to the results and conclusions of this research. Nevertheless, the counselling activity and intervention programs to raise awareness of healthy dietary pattern, including MD, and physical activity impact on maternal–foetal outcomes in women before and during pregnancy is recommended. The midwife role is ideally placed to provide nutrition advice and discuss physical activity changes during pregnancy for primary prevention and best outcomes for both mother and child [34]. Counselling activity is particularly important in women with risk factors, however, there is no guarantee that apparently healthy women follow a healthy diet adequate to their “new” requirements, proper for foetal growth. Finally, in our opinion, to be effective, lifestyle counselling must be anticipated during the preconception period.

## Figures and Tables

**Table 1 nutrients-13-01941-t001:** Characteristics of the study population.

	Total		SGA		Non-SGA		*p*-Value
	(*n* = 100)		(*n* = 10)		(*n* = 90)		(SGA vs. Non-SGA)
	Median	25–75 P	Median	25–75 P	Median	25–75 P	
Age (years)	33	29.00–36.00	30	26.00–36.00	33	30.00–37.00	0.11
Height (cm)	165	160.00–169.50	165	159.00–170.00	165	160.00–169.00	0.7
Weight before pregnancy (Kg)	61.5	54.00–68.50	54	52.00–84.00	62	55.00–68.00	0.73
Pre-gestational BMI (Kg/m^2^)	22.45	16.65–26.50	21	18.80–33.20	22.85	19.80–26.40	0.77
Gestational weight gain (Kg)	13	10.00–16.00	13	10.00–16.00	13	10.00–16.00	0.95
Parity	0	0.00–1.00	1	1.00–1.00	0	0.00–1.00	0.03
MEDI-LITE score	11	9.00–13.00	9	9.00–13.00	11	9.00–13.00	0.28
METs	1815	825.00–3390.00	2040	1260.00–2490.00	1785	810.00–3480.00	0.8
	*n*	%	*n*	%	*n*	%	*p*-value
							(SGA vs. non-SGA)
Nationality							
Italian	81	81	7	70	74	82.23	0.35
non-Italian	19	19	3	30	16	17.77	
Mediterranean basin culture	90	90	8	80	82	91.11	
yes	10	10	2	20	8	8.82	0.27
no							
Level of Education	26	26	3	30	23	25.55	0.76
primary school	41	41	4	40	37	41.11	
high school	33	33	3	30	30	33.34	
university							
Urinary or genital tract infections							0.06
yes	25	25	5	50	20	22.22	
no	75	75	5	50	70	77.78	
Pregnancy-related diseases *							0.01
no diseases	83	84.69	6	60	77	87.5	
gestational diabetes	10	10.21	2	20	8	9.1	
pregnancy-induced hypertension	5	5.1	2	20	3	3.4	
Delivery							0.7
spontaneous	82	82	9	90	73	81.11	
vaginal operative delivery	9	9	0	0	9	10	
C-section	9	9	1	10	8	8.89	
Mediterranean Diet adherence							0.02
MEDI-LITE score > 9	71	71	4	40	67	74.44	
MEDI-LITE score ≤ 9	29	29	6	60	23	25.56	
Level of physical activity							0.48
sedentary (METs < 699)	22	22	2	20	20	22.22	
moderate (METs ranging 700–2519)	40	40	6	60	34	37.78	
high (METs > 2520)	38	38	2	20	36	40	
Alcohol consumption							0.36
never consumed	48	48	6	60	42	46.67	
stopped during pregnancy	33	33	3	30	30	33.33	
no stopped during pregnancy	19	19	1	10	18	20	
Smoking habit							0.45
never smokers	68	68	6	60	62	68.9	
stopped during pregnancy	20	20	2	20	18	20	
no stopped during pregnancy	12	12	2	20	10	11.1	

Legend. SGA: small-for-gestational-age; BMI: Body Mass Index; METs: Metabolic equivalent of task (MET-min) per week; * missing data for two women (“non-SGA” group). For age, height, weight before pregnancy, pre-gestational BMI, gestational weight gain, parity, MEDI-LITE score and METs data were described as median and InterQuartile Range (IQR). For nationality (“Italian; “non-Italian”), Mediterranean basin culture (“yes”; “no”), level of education (“primary school”; “high school”; “university”), urinary or genital tract infections during pregnancy (“yes”; “no”), pregnancy-related diseases (“no diseases”, “gestational diabetes”, “pregnancy-induced hypertension”), delivery (“spontaneous”; “vaginal operative delivery”; “C-section”), Mediterranean Diet adherence (“MEDI-LITE score > 9”; “MEDI-LITE score ≤ 9”), level of physical activity (“sedentary”; “moderate”; “high”), alcohol consumption (“never consumed”; “stopped during pregnancy”; “no stopped during pregnancy”) and smoking habit (“never smokers”; “stopped during pregnancy”; “no stopped during pregnancy”) data are described as absolute number (*n*) and relative frequency (%) of subjects. Significance: *p* < 0.05; *t*-test analysis, Mann–Whitney analysis, chi-square analysis.

**Table 2 nutrients-13-01941-t002:** Correlation between demographic, clinical and lifestyle variables in the overall sample (*n* = 100); data are presented as Spearman’s correlation coefficients and *p* values.

	Level of Education	Height	Weight	Pre-Pregnancy BMI	Gestational Weight Gain	Parity	MEDI-LITE Score	Alcohol Consumption	Smoking Habit	METs
Age	0.30	0.09	0.06	0.05	−0.19	0.18	0.22	0.19	−0.02	−0.06
	0.00	0.38	0.54	0.62	0.06	0.07	0.02	0.06	0.84	0.53
Level of education		0.09	−0.18	−0.22	−0.06	0.04	0.09	0.13	−0.09	−0.06
		0.39	0.07	0.03	0.56	0.67	0.35	0.19	0.38	0.57
Height			0.39	0.04	0.14	0.16	−0.11	−0.17	0.07	0.02
			0.00	0.69	0.16	0.10	0.26	0.08	0.50	0.88
Weight				0.91	−0.02	0.07	−0.18	0.07	0.12	0.03
				<0.0001	0.83	0.49	0.07	0.46	0.25	0.74
Pre-pregnancy BMI					−0.07	0.01	−0.15	0.10	0.11	0.08
					0.51	0.96	0.13	0.33	0.28	0.43
Gestational weight gain						−0.01	−0.17	−0.20	0.12	0.10
						0.94	0.11	0.05	0.25	0.33
Parity							−0.08	0.00	0.02	0.22
							0.45	0.98	0.87	0.03
MEDI-LITE score								−0.01	−0.10	0.06
								0.90	0.35	0.57
Alcohol consumption									0.03	0.03
									0.80	0.77
Smoking habit										0.11
										0.29

Significance: *p* < 0.05; Spearman’s rank correlation. Legend. BMI: Body Mass Index; METs: Metabolic equivalent of task (MET-min) per week.

**Table 3 nutrients-13-01941-t003:** Logistic regression analysis. SGA was the dependent variable while maternal demographic, clinical and lifestyle variables were the independent ones.

Variable	β (SE)	OR	95% CI	*p*-Value
Age	−0.07 (0.07)	0.93	0.81–1.06	0.29
Pre-gestational BMI	0.01 (0.07)	1.01	0.88–1.16	0.88
Gestational weight gain	−0.05 (0.08)	0.95	0.82–1.10	0.66
Pregnancy-related diseases	1.10 (0.56)	3.01	1.01–8.95	0.05
MD adherence	−1.23 (0.77)	0.29	0.06–1,34	0.11
METs	−0.00 (0.00)	0.99	0.99–1.00	0.33
Smoking habit	0.35 (0.13)	1.41	0.52–3.86	0.50
Alcohol consumption	−0.46 (0.65)	0.63	0.18–2.25	0.48

Significance: *p* < 0.05. Legend. BMI: Body Mass Index; OR: Odds Ratio; 95% CI: 95% Confidence Interval.

## Data Availability

All data presented in this study, not yet publicly archived, shall be made available through the corresponding author on request.

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
