# Peer review of "Mediterranean Diet and Lifestyle Habits during Pregnancy: Is There an Association with Small for Gestational Age Infants? An Italian Single Centre Experience"

_nutrients, 2021, doi:10.3390/nu13061941_

Round 1

Reviewer 1 Report

The manuscript present an interesting study. But i have some concerns.

Major concerns:

There is no safe time to drink alcohol during pregnancy. Alcohol can cause problems for the developing baby throughout pregnancy, including before a woman knows she is pregnant. - it has been the case of alcohol consuption in the group entering the study?! if yes, would be interesting to know the age and the level of education of the women. Besides, the reason consuming alcohol (adiction or other reasons).

Besides, drinking alcohol during pregnancy can cause miscarriage, stillbirth, and a range of lifelong physical, behavioral, and intellectual disabilities. - how than could have influence the delivery? this aspect it has been not discussed.

Lines 238-240: women consider it important to improve their lifestyle during reproductive age and pregnancy and in our sample,this has been demonstrated by the high adherence to the MD - from a point of view an imrpovement would be as well quiting consuming alcohol....has been happened in this study?

What about human genetic factors? Any relevance and any relation? Have been asked or tested before/during pregnancy?

Author Response

The authors thank the three reviewers for their helpful comments and suggestions. The answers to the Reviewers’ comments are provided below (in bold the Reviewers’ comments -R-and in italic our answers -A-). Changes in the text were highlighted by the "Track Changes" function in Microsoft Word and highlighted in yellow so that they are easily visible to the Editors and Reviewers.

R. There is no safe time to drink alcohol during pregnancy. Alcohol can cause problems for the developing baby throughout pregnancy, including before a woman knows she is pregnant. - it has been the case of alcohol consuption in the group entering the study?! if yes, would be interesting to know the age and the level of education of the women. Besides, the reason consuming alcohol (adiction or other reasons).

A. The authors agree with the reviewer’s comment. They explored the correlation between alcohol consumption, age and level of education by using Spearman’s rank correlation, as reported in table 2. No significant correlations were found (alcohol consumption-age: coefficient=0.19, p=0.06; alcohol consumption-level of education: coefficient=0.13, p=0.19); for this reason they did not discuss these results.

R. Besides, drinking alcohol during pregnancy can cause miscarriage, stillbirth, and a range of lifelong physical, behavioral, and intellectual disabilities. - how than could have influence the delivery? this aspect it has been not discussed.

A. The authors fully agree with the reviewer's comment. However, they did not investigate how alcohol consumption affected the delivery as the aim of the present research was to retrospectively explore the association between SGA (the dependent variable) and maternal adherence to MD and lifestyle habits (e.g. physicalactivity level, smoking and alcohol consumption), as well as other risk factors for SGA during pregnancy. 

R. Lines 238-240: women consider it important to improve their lifestyle during reproductive age and pregnancy and in our sample,t his has been demonstrated by the high adherence to the MD - from a point of view an improvement would be as well quiting consuming alcohol....has been happened in this study?

A. The authors fully agree with the reviewer's comment. As reported in table 1, alcohol consumption was investigated and women were divided into i) women who never consumed alcohol (48%); ii) women who  stopped alcohol consumption during pregnancy (33%); iii) women who  never stopped alcohol consumption during pregnancy (19%). The authors added a sentence in the materials and methods section to better describe how alcohol consumption has been recorded (please see page 3, lines 132-134).

R. What about human genetic factors? Any relevance and any relation? Have been asked or tested before/during pregnancy?

A.  The authors thank the reviewer for that comment. They are conscious that causes for SGA birth include environmental factors and inherited genetic mutations (Int J Pediatr Endocrinol. 2012 May 15;2012(1):12. doi: 10.1186/1687-9856-2012-12). However this is a a retrospective population-based study (please see page 2, line 73 –materials and methods section) aimed at exploring the association between SGA and maternal adherence to MD and lifestyle habits (e.g. physical activity level, smoking and alcohol consumption). Therefore, since genetic mutations associated with SGA are not assessed in the routine practice and data were evaluated retrospectively, genetic informations were not available. It would be interesting to plan a longitudinal study considering also this important aspect. 

Reviewer 2 Report

The present manuscript entitled “Mediterranean diet and lifestyle habits during pregnancy: is there an association with small for gestational age infants? An Italian single center experience. " deals with the association between SGA, maternal adherence to the Mediterranean diet, lifestyle habits and other risk factors during pregnancy. The study is interesting. Here are some suggestions point by point:
- The introduction is well supported by the authors, in addition to having a good justification and ending with the objective of the study.
- In the material and methods section, I advise adding, in the first paragraph (line 70-78), a subscript, making clear the ethics committee section.
- From lines 90 to 93 there are numerous examples, in a scientific academic text those examples should not appear, all the study variables should be added.
- Line 104 would be missing a reference to support that questionnaire and what is mentioned.
- In the results section, in the 5th line, the initials IQR are not developed the first time they are named.
- In Table 1, the explanations of the table would be better as a footer.
- Table 1 would be better seen vertically, reducing the spaces.
- The discussion section compares the results of the present study with those of other authors.
- The conclusion highlights well the most important results and proposes future interventions and lines of research.
- Check the bibliographic references, reference 7 for example, the year should be in bold.

Author Response

The authors thank the three reviewers for their helpful comments and suggestions. The answers to the Reviewers’ comments are provided below (in bold the Reviewers’ comments -R-and in italic our answers -A-). Changes in the text were highlighted by the "Track Changes" function in Microsoft Word and highlighted in yellow so that they are easily visible to the Editors and Reviewers.

R. The present manuscript entitled “Mediterranean diet and lifestyle habits during pregnancy: is there an association with small for gestational age infants? An Italian single center experience" deals with the association between SGA, maternal adherence to the Mediterranean diet, lifestyle habits and other risk factors during pregnancy. The study is interesting. Here are some suggestions point by point. The introduction is well supported by the authors, in addition to having a good justification and ending with the objective of the study. In the material and methods section, I advise adding, in the first paragraph (line 70-78), a subscript, making clear the ethics committee section.

A. The authors agree on the importance of stating Ethics Committee (EC) approval, indeed they have reported the information regarding the Ethical Committee in the materials and methods section, where EC information belongs as usual in scientific  manuscripts: “The study was approved by the Human Ethics Committee of Fondazione IRCCS Policlinico S. Matteo of Pavia (Protocol number: 20180022618; 6/12/2018) and it was conducted according to the Good Clinical Practice guidelines” (pleas,e see page 2, lines 79-81 - materials and methods section), as well as in the “Institutional Review Board Statement” section at the end of the manuscript.

R. From lines 90 to 93 there are numerous examples, in a scientific academic text those examples should not appear, all the study variables should be added.

A. The authors would like to reassure the reviewer that those in parenthesis are not examples but all the factors collected from the medical records. Referring to “Other maternal and foetal outcomes were also collected from the medical records. In particular, i) diseases during pregnancy (no diseases reported; gestational diabetes mellitus, GDM; pregnancy-induced hypertension); ii) onset of urinary or genital tract infections in the mother during pregnancy (yes; no); iii) type of delivery (sponta-neous delivery; vaginal operative delivery; caesarean section)”: these maternal and fetal outcomes have been collected and considered in the study as also reported in table 1. In the materials and methods section, the authors listed and described all the variables evaluated in the study, as requested by the authors’ guidelines of the journal. R. Line 104 would be missing a reference to support that questionnaire and what is mentioned. A. The authors thank the reviewer and added the reference, as requested (please, see page 3, line 105).

R. In the results section, in the 5th line, the initials IQR are not developed the first time they are named.

A. The acronym IQR was first mentioned in the statistical analysis section (please, see page 3, lines 139-140).

R. In Table 1, the explanations of the table would be better as a footer.

A. The table explanation has been reported as a footer, as suggested.

R. Table 1 would be better seen vertically, reducing the spaces.

A. Authors agree with the reviewer’s suggestion but the table final version will be edited in the final draft, once the article will be sent back as proofs to be revised by the authors for publication  The discussion section compares the results of the present study with those of other authors. The conclusion highlights well the most important results and proposes future interventions and lines of research.

R. Check the bibliographic references, reference 7 for example, the year should be in bold.

A. The authors thank the reviewer and checked all the “references section”, as suggested. 

Reviewer 3 Report

The brief report conducted by Rachele De Giuseppe et al. which aimed to explore retrospectively the association between SGA and the maternal adherence to the MD and lifestyle habits as well as other SGA’s risk factors during pregnancy in a small Italian cohort is well designed and very well written. I have only minor suggestions before it can be considered for publication:

Some directions for future investigations should be provided at the end of the abstract.

Some recent literature about the update of the Mediterranean Diet Pyramid needs to be referenced in the Introduction:

Serra-Majem, L., Tomaino, L., Dernini, S., Berry, E. M., Lairon, D., Ngo de la Cruz, J., ... & Trichopoulou, A. (2020). Updating the Mediterranean diet pyramid towards sustainability: Focus on environmental concerns. International Journal of Environmental Research and Public Health17(23), 8758.

Fernandez, M. L., Raheem, D., Ramos, F., Carrascosa, C., Saraiva, A., & Raposo, A. (2021). Highlights of current dietary guidelines in five continents. International Journal of Environmental Research and Public Health18(6), 2814.

The authors should consider adding a flowchart to section 2, providing all the steps undertaken in the carrying out of the present study.  

Author Response

The authors thank the three reviewers for their helpful comments and suggestions. The answers to the Reviewers’ comments are provided below (in bold the Reviewers’ comments -R-and in italic our answers -A-). Changes in the text were highlighted by the "Track Changes" function in Microsoft Word and highlighted in yellow so that they are easily visible to the Editors and Reviewers.

R. The brief report conducted by Rachele De Giuseppe et al. which aimed to explore retrospectively the association between SGA and the maternal adherence to the MD and lifestyle habits as well as other SGA’s risk factors during pregnancy in a small Italian cohort is well designed and very well written. I have only minor suggestions before it can be considered for publication: Some directions for future investigations should be provided at the end of the abstract.

A. The authors agree with the reviewer’s comment and added a sentence with future directions at the end of the abstract.

R. Some recent literature about the update of the Mediterranean Diet Pyramid needs to be referenced in the Introduction: •    Serra-Majem, L., Tomaino, L., Dernini, S., Berry, E. M., Lairon, D., Ngo de la Cruz, J., ... & Trichopoulou, A. (2020). Updating the Mediterranean diet pyramid towards sustainability: Focus on environmental concerns. International Journal of Environmental Research and Public Health, 17(23), 8758. •    Fernandez, M. L., Raheem, D., Ramos, F., Carrascosa, C., Saraiva, A., & Raposo, A. (2021). Highlights of current dietary guidelines in five continents. International Journal of Environmental Research and Public Health, 18(6), 2814.

A. The authors agree with the reviewer’s comment and added the references suggested in the introduction  section.

R. The authors should consider adding a flowchart to section 2, providing all the steps undertaken in the carrying out of the present study.  

A. The authors  thank the reviewer and added a flowchart as suggested; please refer to figure 1. 

Round 2

Reviewer 1 Report

The authors improved the manuscript and adressed al recommandations/remarks/suggestions. Overall this manuscript merits to be published.

Author Response

The authors thank the #reviewer 1 for his/her suggestions that have been considered already in the first round of revision since  recommended also by the other reviewer. 

In particular:

i) The manuscript was revised by an English native-language speaker.

ii) The authors added recent and relevant literature about the update of the Mediterranean Diet Pyramid in the introduction section:  • Serra-Majem, L., Tomaino, L., Dernini, S., Berry, E. M., Lairon, D., Ngo de la Cruz, J., ... & Trichopoulou, A. (2020). Updating the Mediterranean diet pyramid towards sustainability: Focus on environmental concerns. International Journal of Environmental Research and Public Health, 17(23), 8758. (Reference number 6). • Fernandez, M. L., Raheem, D., Ramos, F., Carrascosa, C., Saraiva, A., & Raposo, A. (2021). Highlights of current dietary guidelines in five continents. International Journal of Environmental Research and Public Health, 18(6), 2814. (Reference number 7).

iii) The authors added a flowchart (figure1) to describe adequately methods, providing all the steps undertaken in the carrying out of the present study.